# 3D Neural Embedding Likelihood for Robust Sim-to-Real Transfer in Inverse Graphics

## Abstract

A central challenge in 3D scene perception via inverse graphics is robustly modeling the gap between 3D graphics and real-world data. We propose a novel 3D Neural Embedding Likelihood (3DNEL) over RGB-D images to address this gap. 3DNEL uses neural embeddings to predict 2D-3D correspondences from RGB and combines this with depth in a principled manner. 3DNEL is trained entirely from synthetic images and generalizes to real-world data. To showcase this capability, we develop a multi-stage inverse graphics pipeline that uses 3DNEL for 6D object pose estimation from real RGB-D images. Our method outperforms the previous state-of-the-art in sim-to-real pose estimation on the YCB-Video dataset, and improves robustness, with significantly fewer large-error predictions. Unlike existing bottom-up, discriminative approaches that are specialized for pose estimation, 3DNEL adopts a probabilistic generative formulation that jointly models multi-object scenes. This generative formulation enables easy extension of 3DNEL to additional tasks like object and camera tracking from video, using principled inference in the same probabilistic model without task specific retraining.

## 1 Introduction

There is a widespread need for models that bridge the gap between 3D graphics and real RGB-D data. Accurate simulation environments exist in domains such as autonomous driving, augmented reality, and robotic manipulation, yet robust 3D scene perception remains a central bottleneck. "Inverse graphics" is an appealing approach to 3D scene understanding that treats scene perception as the inverse problem to 3D graphics. However, in practice these methods have been outperformed by more bottom-up, discriminative approaches, especially those using deep learning (LeCun et al., 2015). A key challenge in inverse graphics is modeling the gap between rendered images and observed real-world images. This paper aims to address this gap with 3D Neural Embedding Likelihood (3DNEL), a likelihood model of RGB-D images that is trained entirely from synthetic data and generalizes to real-world data. 3DNEL uses learned neural embeddings to predict dense 2D-3D correspondences from RGB and combines this with 3D information from depth in a principled way.

To showcase 3DNEL's capabilities in sim-to-real transfer, we develop a multi-stage inverse graphics pipeline (MSIGP) that uses 3DNEL for 6D object pose estimation. We demonstrate that 3DNEL can be applied to real RGB-D images without having to train on any real data. Our 3DNEL MSIGP consists of (1) a coarse enumerative procedure that generates pose hypotheses and an initial estimate of the 3D scene and (2) an iterative Markov Chain Monte Carlo (MCMC) process that finetunes the 3D scene. We empirically evaluate 3DNEL MSIGP on the popular YCB-Video (YCB-V) dataset (Xiang et al., 2018). 3DNEL MSIGP outperforms the previous state-of-the-art (SOTA) SurfEMB (Haugaard & Buch, 2022) in sim-to-real 6D pose estimation, albeit at the cost of increased computation. It is also significantly more robust: we show over 50% reduction in high-error pose predictions compared to SurfEMB. Extensive ablation studies illustrate the source of performance improvements.

Existing approaches for 6D pose estimation are predominantly discriminative and bottom-up, and are specialized to the specific task of 6D pose estimation. In contrast, 3DNEL adopts a probabilistic generative formulation which extends beyond just pose estimation. To demonstrate the value of 3DNEL's generative formulation, we present additional experiments on 3DNEL's easy extension to object and camera tracking from video, using principled inference in the same probabilistic model without task specific retraining.

## 2    RELATED WORK

**3D Inverse Graphics**    Our method follows a long line of work in the "analysis-by-synthesis" paradigm that treats perception as the inverse problem to computer graphics (Kersten & Yuille, 1996; Yuille & Kersten, 2006; Lee & Mumford, 2003; Kersten et al., 2004; Mansinghka et al., 2013; Kulkarni et al., 2015). While elegant and conceptually appealing, robustly modeling the gap between 3D graphics and real-world data, especially using appearance information, remains a central challenge in 3D inverse graphics. Our key observation is that, dense 2D-3D correspondences, widely used in many recent 6D pose estimation methods (Hodan et al., 2020; Li et al., 2019; He et al., 2020; Tremblay et al., 2018; Florence et al., 2018; Haugaard & Buch, 2022), provide a natural way to model appearance information in 3D inverse graphics, and can be combined with depth information into a unified probabilistic generative model to effectively bridge the sim-to-real gap.

**Sim-to-Real Transfer**    Most computer vision systems require annotated real-world data for training (Hodan et al., 2017; Brachmann et al., 2014; Xiang et al., 2018; Krizhevsky et al., 2017) in order to achieve strong performance on real-world data at test time, since without it the "sim-to-real gap" is too difficult to overcome. Practically, it is tedious and expensive to collect and annotate real data. Recent advances in photorealistic rendering and physics-based simulations (Hodan et al., 2018; Tremblay et al., 2018; Denninger et al., 2019) have resulted in strong performance of models trained entirely on synthetic data. SurfEMB (Haugaard & Buch, 2022) is one such model that obtains SOTA performance in 6D pose estimation, outperforming numerous approaches that use real data for training. 3DNEL builds upon SurfEMB's learned dense 2D-3D correspondences, and expands previous work (Gothoskar et al., 2021) on probabilistic modeling of real depth data. It combines RGB and depth information into a unified probabilistic generative model, and improves SurfEMB nontrivially in both accuracy and robustness to achieve new SOTA in sim-to-real 6D pose estimation.

**6D Object Pose Estimation**    6D object pose estimation aims to infer the rigid $\mathbb{SE}(3)$ transformation (position and orientation) of an object in the camera frame, given an image observation. Discriminatively trained deep learning approaches (Xiang et al., 2018; Li et al., 2018; Deng et al., 2019; He et al., 2020; Sundermeyer et al., 2018) have, in general, outperformed more traditional feature or template matching methods (Besl & McKay, 1992; Rusinkiewicz & Levoy, 2001; Lowe, 1999; Rothganger et al., 2006; Collet et al., 2011). With the advent of RGB-D cameras, there has been a growing interest in leveraging both the appearance information from RGB and the geometric shape information from depth. Existing methods either use depth to post process estimations from RGB (Xiang et al., 2018; Haugaard & Buch, 2022), or fuse learned features from both RGB and real, noisy depth (Wang et al., 2019; He et al., 2021). 3DNEL differs from prior work in two important ways. (1) 3DNEL combines RGB and depth information in a principled probabilistic model, which enables superior sim-to-real transfer. (2) 3DNEL jointly models multi-object scenes using a probabilistic generative formulation, allowing easy extension to tasks beyond 6D pose estimation.

## 3    METHODS

### 3.1    PRELIMINARIES

**Likelihood for 3D Inverse Graphics**    3D inverse graphics formulates the perception problem as searching for the 3D scene description that can be rendered by a graphics engine to best reconstruct the input image. A central challenge in applying the 3D inverse graphics approach to real images is robustly modeling the "gap" between rendered and real images. In this paper we aim to develop a likelihood $\mathbb{P}(\textit{Observed RGB-D Image}|\textit{3D scene description})$ that can combine shape and appearance information in a principled way to robustly assess how well an observed RGB-D image is explained by a 3D scene description. A naive approach is to render the 3D scene description to an RGB-D image and define the likelihood as a noise model that directly compares the rendered and real RGB-D images. Recent work 3DP3 (Gothoskar et al., 2021) demonstrates promising performance of this approach when applied to depth images. However, it is much more challenging to specify a sensible noise model operating directly on RGB images. Intuitively, the "gap" between rendered and real depth images is mainly due to small spatial displacements, yet the "gap" between rendered and real RGB images comes from a variety of different factors. In addition, a principled combination of RGB and depth information in a unified likelihood remains an open problem.

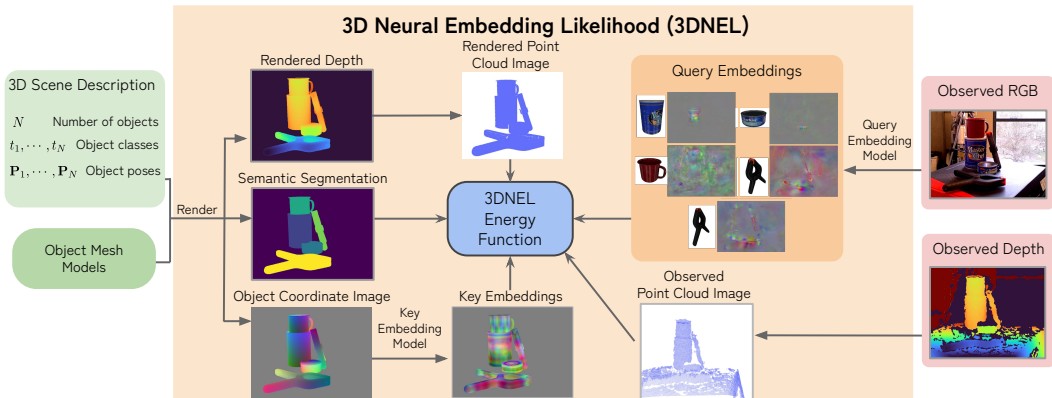

Figure 1: **Evaluating 3DNEL** 3DNEL defines the probability of an observed RGB-D image conditioned on a 3D scene description. We first render the 3D scene description into: (1) a depth image, which is transformed to a rendered point cloud image, (2) a semantic segmentation map, and (3) the object coordinate image (each pixel contains the object frame coordinates of the object surface point from which the pixel originates). The object coordinate image is transformed, via the key models, into key embeddings. The observed RGB image is transformed, via the query models, into query embeddings. The observed depth is transformed into an observed point cloud image. The 3DNEL Energy Function (Equation 1) is evaluated using the rendered point cloud image, semantic segmentation, key embeddings, the observed point cloud image, and query embeddings.

**3D Scene Description** We describe 3D scenes in terms of the objects in the scene and their poses. Each object is associated with a textured mesh, which captures the 3D shape and appearance information of the object. The poses of the objects in the scene are represented in $\mathbb{SE}(3)$ (rigid transformations in 3D) which define an object's position and orientation in space. The 3D scene description is specified as the number $N$ of objects in the scene, their classes $t_1, \cdots, t_N \in \{1, \cdots, M\}$, and corresponding poses $\mathbf{P}_1, \cdots, \mathbf{P}_N \in \mathbb{SE}(3)$. We assume uniform prior distributions over object poses (uniform over a bounded volume for position and uniform on $\mathbb{SO}(3)$ for orientation).

**Noise Model on Depth Information** We use the probabilistic model $\mathbb{P}_{depth}(\mathbf{c}|\tilde{\mathbf{c}}; r) = \frac{\mathbf{1}[||\mathbf{c}-\tilde{\mathbf{c}}||_2 \leq r]}{\frac{4}{3}\pi r^3}$ from 3DP3 as our noise model on depth information. $\mathbb{P}_{depth}$ is a uniform distribution in a radius-$r$ ball centered at a rendered point $\tilde{\mathbf{c}} \in \mathbb{R}^3$, and models the small spatial displacements in the observed point $\mathbf{c} \in \mathbb{R}^3$. $r$ is a hyperparameter that controls the variance of the noise model.

**Dense 2D-3D Correspondence Distributions as Noise Models on RGB Information** Instead of directly operating on RGB images, we leverage learned dense 2D-3D correspondences to specify the noise model on RGB information. Concretely, we build on SurfEMB (Haugaard & Buch (2022)) and use the proposed dense 2D-3D correspondence distributions over object surfaces. For each object class $t \in \{1, \cdots, M\}$, SurfEMB learns two neural embedding models: (1) a *query embedding model* which maps an RGB image $\mathbf{I} \in \{0, \cdots, 255\}^{H \times W \times 3}$ to a set of query embeddings $\mathbf{Q}^t \in \mathbb{R}^{H \times W \times 3}$, one for each 2D pixel location, and (2) a *key embedding model* $g_t : \mathbb{R}^3 \mapsto \mathbb{R}^E$ which maps each 3D location $\mathbf{x} \in \mathbb{R}^3$ (coordinate in the object frame) on the object surface to a key embedding $g_t(\mathbf{x}) \in \mathbb{R}^E$. Given a pixel with query embedding $\mathbf{q} \in \mathbb{R}^E$, SurfEMB defines a surface distribution $\mathbb{P}_{RGB}(\mathbf{x}|\mathbf{q}, t) \propto \exp(\mathbf{q}^T g_t(\mathbf{x}))$ describing which point $\mathbf{x}$ on the object surface the given pixel corresponds to. Importantly, these models can be trained entirely from synthetic data (with photorealistic rendering and physics-based simulations). See Appendix A.1 for a more detailed review of the query and key embedding models and the dense 2D-3D correspondence distributions.

## 3.2 3D Neural Embedding Likelihood (3DNEL)

**Processing 3D Scene Description for 3DNEL evaluation** For a given 3D scene description, we use a graphics engine to render it into: (1) A rendered point cloud image $\tilde{\mathbf{C}} \in \mathbb{R}^{H \times W \times 3}$, where $\tilde{\mathbf{C}}_{i,j} \in \mathbb{R}^3$ represents the camera frame coordinate at pixel $(i, j)$. (2) A semantic segmentation map $\tilde{\mathbf{S}} \in \{0, 1, \cdots, M\}^{H \times W}$ where $\tilde{\mathbf{S}}_{i,j}$ represents the class to which the pixel $(i, j)$ belongs. Here 0

Figure 2: **Using 3DNEL for 3D Scene Parsing** The 3DNEL MSIGP pipeline starts by computing the query embeddings for each object from the observed RGB, and the observed point cloud image from the depth. Then, a fast enumerative procedure produces the pose hypotheses for the objects, and construct an initial 3D scene description. We further perform MCMC finetuning of the 3D scene description with 3DNEL using three types of MH proposals (1) pose hypotheses proposals (2) ICP proposals to align an object to point cloud data, and (3) random walk proposals that refines poses with local perturbations. The result is a 3D scene description that explains the observed RGB-D image. 3DNEL's joint modeling of multiple objects through the mixture model formulation enables robust estimation on this challenging scene with two similar-looking clamps, while SurfEMB (bottom left) makes per-object predictions and incorrectly predicts both clamps to be in the back.

represents background. (3) An object coordinate image $\tilde{\mathbf{X}} \in \mathbb{R}^{H \times W \times 3}$ where $\tilde{\mathbf{X}}_{i,j}$ represents the coordinate at pixel $(i,j)$ in the object frame of the object of class $\tilde{\mathbf{S}}_{i,j}$.

**Processing Observed RGB-D Image for 3DNEL Evaluation**    For an observed RGB image $\mathbf{I} \in \{0, \cdots, 255\}^{H \times W \times 3}$ and depth image, we use the learned query embedding models to obtain $M$ sets of query embeddings $\mathbf{Q}^t \in \mathbb{R}^{H \times W \times E}, t \in \{1, \cdots, M\}$, one for each object class, and use camera intrinsics to unproject the depth image into an observed point cloud image $\mathbf{C} \in \mathbb{R}^{H \times W \times 3}$.

**3DNEL evaluation**    Figure 1 visualizes 3DNEL evaluation using processed 3D scene descriptions and observed RGB-D images. 3DNEL combines the noise model $\mathbb{P}_{depth}$ on depth information from 3DP3 and the dense 2D-3D correspondence distributions $\mathbb{P}_{RGB}$ from SurfEMB into a unified likelihood for sim-to-real transfer. 3DNEL jointly models multiple objects in a scene through a mixture model formulation, resulting in a probabilistic generative model of real RGB-D images.

Intuitively, we assess how well each pixel $(i,j)$ in the observed point cloud image $\mathbf{C}$ is explained by a pixel $(\tilde{i}, \tilde{j})$ in the rendered point cloud image $\tilde{\mathbf{C}}$, by combining the noise model $\mathbb{P}_{depth}$ on depth and the noise model $\mathbb{P}_{RGB}$ on RGB. To jointly model multiple objects in a scene, we assume each pixel $(i,j)$ in $\mathbf{C}$ can be explained by multiple pixels in $\tilde{\mathbf{C}}$.

We formalize this with a mixture model formulation, where the mixture component associated with the rendered pixel $(\tilde{i}, \tilde{j})$ combines $\mathbb{P}_{depth}$ and $\mathbb{P}_{RGB}$ to assess how well the observed pixel $(i,j)$ is explained by the rendered pixel $(\tilde{i}, \tilde{j})$. To model background pixels in $\mathbf{C}$, we assume the observed point cloud image $\mathbf{C}$ resides in a bounded region of volume $B$, and introduce a uniform distribution $\mathbb{P}_{background}(\mathbf{c}; B) = 1/B$ on the bounded region with mixture probability $\epsilon$ as an additional mixture component for background modeling. Representing the total number of non-background pixels in the rendered images as $\tilde{K} = \sum_{\tilde{i}, \tilde{j}} \mathbf{1}[\tilde{\mathbf{S}}_{\tilde{i}, \tilde{j}} > 0]$, the mixture probability for the mixture component

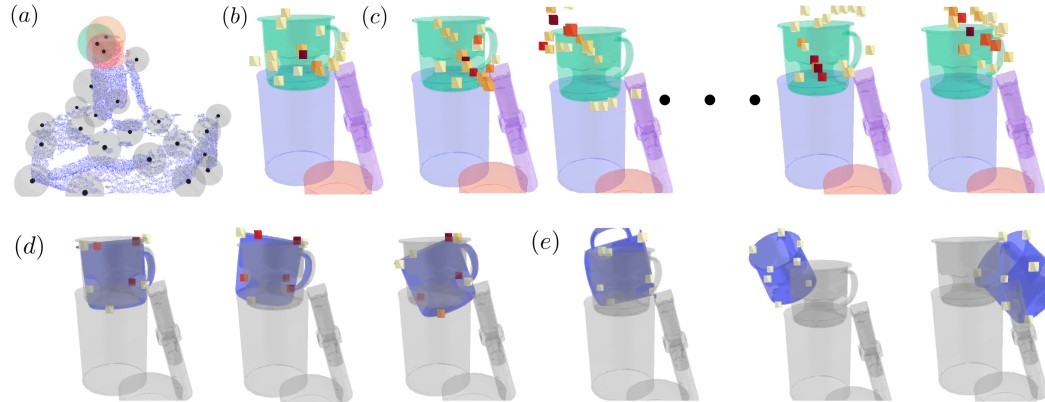

Figure 3: **Enumeration-based Pose Hypotheses Generation** We visualize our enumeration-based pose hypotheses generation process using a mug object as an example. **Figure 3(a)** visualizes spherical voting for identifying the mug center. The red points in the observed point cloud represent points associated with the mug. The 3 colored spheres illustrate how votes from points on the mug combine to identify the mug center. The same procedure can also be used to identify the $n_k$ keypoints on the mug surface. **Figure 3(b)** visualizes 20 top-scoring voxels from the voxel grid associated with the mug center. **Figure 3(c)** visualizes 20 top-scoring voxels from the voxel grids associated with the $n_k$ keypoints. **Figures 3(d)(e)** show how we score pose hypotheses by summing the scores of the voxels the corresponding $n_k$ keypoints fall into. **Figure 3(d)** shows top-scoring pose hypotheses. **Figure 3(e)** shows how pose hypotheses far away from ground truth get low scores. In **Figures 3(b) to (e)**, light yellow represents low voxel scores, while dark red represents high voxel scores.

associated with rendered pixel $(\tilde{i}, \tilde{j})$ is given by $(1 - \epsilon)/\tilde{K}$. Since the query embedding at a pixel depends on the entire image $\mathbf{I}$, the mixture components are not properly normalized. This leads to the following energy-based formulation for 3DNEL:

$$\mathbb{P}(\mathbf{I}, \mathbf{C}|\tilde{\mathbf{C}}, \tilde{\mathbf{S}}, \tilde{\mathbf{X}}) \propto \prod_{\mathbf{c}} \left( \epsilon \mathbb{P}_{background}(\mathbf{c}; B) + \frac{1-\epsilon}{\tilde{K}} \sum_{\tilde{\mathbf{c}}:\tilde{s}>0} \mathbb{P}_{depth}(\mathbf{c}|\tilde{\mathbf{c}}; r) \mathbb{P}_{RGB}(\tilde{\mathbf{x}}|\mathbf{q}^{\tilde{s}}, \tilde{s}) \right) \quad (1)$$

where we denote $\mathbf{C}_{i,j}$ by $\mathbf{c}$, $\tilde{\mathbf{C}}_{\tilde{i},\tilde{j}}$ by $\tilde{\mathbf{c}}$, $\tilde{\mathbf{S}}_{\tilde{i},\tilde{j}}$ by $\tilde{s}$, $\tilde{\mathbf{X}}_{\tilde{i},\tilde{j}}$ by $\tilde{x}$, and $\mathbf{Q}_{i,j}^{t}$ by $\mathbf{q}^{t}$. The product is over all observed pixels, and the sum is over all non-background rendered pixels. $\epsilon$, $B$ and $r$ are hyper-parameters that we pick in the experiments. See Appendix A.2 for more details.

## 3.3 Inferring the 3D scene description from RGB-D

Figure 2 gives an overview of 3DNEL MSIGP for 6D object pose estimation.

**Enumeration-based Pose Hypotheses Generation** We develop a novel spherical voting procedure and a heuristic scoring using the query embeddings and observed point cloud image $\mathbf{C}$ defined in Section 3.2, and use them in an enumerative procedure to efficiently generate pose hypotheses. We use the object center and $n_k$ points sampled using farthest point sampling from the object surface as our keypoints, and discretize the camera frame space into a $L_x \times L_y \times L_z$ voxel grid .

For a given keypoint, our spherical voting procedure aggregates information from the entire image to score how likely the keypoint is present at different voxel locations, and stores the scores in a voxel grid. Figure 3(a) visualizes spherical voting for the center $x^*$ of the mug object: for pixel location $(i, j)$ with camera frame coordinate $\mathbf{c} \in \mathbb{R}^3$ and query embedding $\mathbf{q} \in \mathbb{R}^E$, we identify its most likely corresponding point on the mug surface $x = \arg\max_{\tilde{x}} \mathbb{P}_{RGB}(\tilde{x}|\mathbf{q}, t)$, calculate the distance $r_x = ||x - x^*||_2$ from $x$ to $x^*$ in the object frame, and cast votes (Qi et al., 2019) with weight $p_{i,j} = \max_{\tilde{x}} \mathbb{P}_{RGB}(\tilde{x}|\mathbf{q}, t)$ towards all points on a sphere of radius $r_x$ centered at $\mathbf{c}$. Figure 3(b) visualizes the 20 top-scoring voxels from the voxel grid for the mug center, and Figure 3(c) visualizes the 20 top-scoring voxels from the voxel grids for the $n_k$ keypoints on the mug surface.

We coarsely discretize the object pose space. We reuse the same camera frame space discretization into a voxel grid, and use the $L_x \times L_y \times L_z$ voxel centers to discretize the location space. We use

a customized procedure (Appendix A.5) to generate $n_r$ rotations and discretize the rotation space. We use the voxel grid for the object center to identify top-scoring object locations, and score all $n_r$ rotations at these locations. We score a given object pose with the sum of the scores of the voxels the corresponding $n_k$ keypoints fall into. Figure 3(d) visualizes 3 example top pose hypotheses from the enumerative procedure. Figure 3(e) visualizes how poses far away from ground truth get low scores from our heuristic scoring. See Appendix A.3 for a formal description of the process.

**MCMC Finetuning** We generate an initial 3D scene description using top scoring pose hypothesis for each object instance in the scene based on the pose hypotheses for its class, and use MCMC, in particular the Metropolis-Hastings (MH) algorithm, to iteratively finetune the 3D scene description. For a scene with $N$ objects, at each iteration, MH proposes to change the pose $P_i$ of a single object $i$, and accepts the proposal with the MH acceptance probability. See Figure 2 for visualizations.

We use three different proposals, all of which sample a pose's position from a spherical normal distribution centered at a desired position and the pose's orientation from a von Mises-Fisher distribution centered at a desired orientation. (1) Pose hypotheses proposals go through the $N$ objects multiple times. For object $i$ of class $t_i$, the pose hypotheses proposals sequentially go through the generated pose hypotheses for object class $t_i$, sample a pose centered at each of the pose hypotheses, and propose to change the pose of object $i$ to the sampled pose. (2) ICP proposals iterate over all objects and use ICP to align each object's 3D model to the observed 3D point cloud with the point cloud from all the other objects masked out, and make a proposal centered at the resulting pose. (3) Finally, random walk proposals that randomly perturb object poses. See Appendix A.4 for details.

## 4 EXPERIMENTS

We conduct extensive experiments on YCB-V, a popular benchmark for 6D object pose estimation. We establish that 3DNEL MSIGP significantly improves the previous SOTA SurfEMB in sim-to-real 6D object pose estimation, and is more robust in terms of reducing large pose estimation errors and overcoming failures of 2D detectors. Detailed ablations illustrate the source of performance improvements, and additional experiments on easy extension of 3DNEL to object and camera tracking from video demonstrates the value of 3DNEL's formulation as a probabilistic generative model.

**Evaluation** We follow the evaluation protocol of the Benchmark for 6D Object Pose Estimation (BOP) challenge (Hodan et al., 2018). The task is to estimate the 6D poses of objects in a scene, from a single RGB-D image. The BOP challenge assumes knowledge of the number of instances of each object class in the scene. For a predicted pose, we calculate three error metrics with respect to the ground truth pose: Visible Surface Discrepancy (VSD) (Hodaň et al., 2016; Hodan et al., 2018), Maximum Symmetry-Aware Surface Distance (MSSD) (Drost et al., 2017), and Maximum Symmetry-Aware Projection Distance (Brachmann et al., 2016). Average recalls $AR_{VSD}$, $AR_{MSSD}$, $AR_{MSPD}$ are computed for each error metric across a range of error thresholds. The aggregate Average Recall (as reported in Table 1) is the average of $AR_{VSD}$, $AR_{MSSD}$, and $AR_{MSPD}$.

**Training** To allow easy comparison with baselines, we use publicly released pretrained SurfEMB models for our experiments. These include query models, key models, mask predictors for different object classes, and an additional 2D detector from CosyPose (Labbé et al., 2020), all trained entirely from synthetic data. While sharing the same underlying models, our 3DNEL MSIGP is significantly different from SurfEMB's PnP+RANSAC procedure, and demonstrates nontrivial improvements in overall accuracy as well as robustness. We expect additional training tailored towards the 3DNEL formulation can further improve performance, but leave this to future work.

**3DNEL MSIGP** We follow He et al. (2021) and use Ku et al. (2018) to fill in missing depth. Recent strong pose estimation methods commonly rely on 2D detection and mask prediction. Since our spherical voting procedure can robustly generate pose hypotheses by aggregating information from the entire image, 3DNEL MSIGP can work well even without separate 2D detection and mask prediction (Table 1, *3DNEL MSIGP (No 2D Detection)*) while SurfEMB completely fails without these. However, empirically we find that we can leverage additional 2D detection and mask prediction to allow spherical voting to aggregate information only from regions of the query embedding images that are likely relevant to the objects, and significantly outperform SurfEMB using identical information. We pick hyperparameters by visually inspecting inference results on a small number of real training images outside the test set. See Appendix A.5 for details.

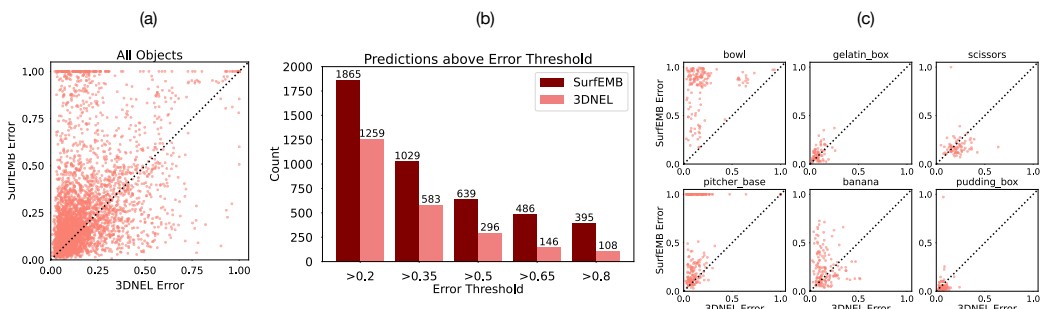

Figure 4: **3DNEL MSIGP yields greater robustness than SurfEMB** (a) Comparison of prediction error (measured by VSD) between SurfEMB and 3DNEL MSIGP across 4123 object instances in YCB-V. Each instance is represented as a point on the scatter plot. Points above the dashed line represent instances for which 3DNEL MSIGP has lower prediction error. (b) The number of instances with prediction error above a certain error threshold, across multiple thresholds. 3DNEL MSIGP makes significantly less high-error pose predictions than SurfEMB (over 50% less above 0.5). (c) Scatter plots per object classes for 6 representative object classes out of the 21 object classes.

| Category | Method | Average Recall |
|---|---|---|
| **Sim-to-Real** | 3DNEL MSIGP (Ours) | **83.68%** |
| | SurfEMB (Haugaard & Buch, 2022) | 80.00% |
| | FFB6D (He et al., 2021) | 75.80% |
| **Ablations** | 3DNEL MSIGP (No RGB in Likelihood) | 61.57% |
| | 3DNEL MSIGP (No Depth in Likelihood) | 50.85% |
| | 3DNEL MSIGP (SurfEMB initialization + MCMC) | 82.73% |
| | 3DNEL MSIGP (Initialization only) | 62.36% |
| | 3DNEL MSIGP (No 2D Detection) | 76.53% |
| **Real and Synthetic Training** | FFB6D ((He et al., 2021) | 85.50% |
| | CosyPose (Labbé et al., 2020) (4 Views) | 84.00% |
| | SurfEMB (Haugaard & Buch, 2022) | 82.40% |
| | Pix2Pose (Park et al., 2019) | 78.00% |
| | W-PoseNet (Xu et al., 2019) | 77.90% |
| | Koenig-Hybrid (König & Drost, 2020) | 70.10% |

Table 1: **3DNEL MSIGP yields greater accuracy than baselines and ablations** We report Average Recall on the YCB-Video dataset. 3DNEL MSIGP is the top-performing sim-to-real method for 6D pose estimation. Results are averaged over 5 runs. Standard deviation is below $0.2\%$ for all setups.

Our superior performance comes at the cost of increased computation. Empirically we observe our current (highly unoptimized) implementation can be 6x slower than SurfEMB (20 to 30s on static images, 15 to 20s for tracking, on a machine with a single NVIDIA A100 GPU). In Section 5 we discuss this limitation and propose some potential solutions.

## 4.1 ACCURACY

In Table 1, we report the Average Recall for 3DNEL MSIGP, its various ablations and representative bottom-up baselines. In the first section we compare with two baselines trained entirely on synthetic data: the current SOTA SurfEMB (Haugaard & Buch, 2022) which uses depth to heuristically refine pose estimates with RGB information, and FFB6D (He et al., 2021) which is a bottom-up approach that fuses appearance information from RGB with geometry information from depth. 3DNEL formulates a probabilistic 3D likelihood over depth data and appearance information from RGB in the form of 2D-3D correspondence distributions, and significantly outperforms SurfEMB, FFB6D.

In the second section, we evaluate various ablations of 3DNEL MSIGP. We first investigate the importance of principled combination of RGB and depth. We evaluate *3DNEL MSIGP (No RGB in Likelihood)*, where we drop $\mathbb{P}_{RGB}$ in Equation 1, and *3DNEL MSIGP (No Depth in Likelihood)* where we replace the 3D ball $\mathbf{1}[||\mathbf{C}_{i,j} - \tilde{\mathbf{C}}_{\tilde{i},\tilde{j}}||_2 \leq r]$ in Equation 1 with a $3 \times 3$ 2D patch. Both are substantially worse. We next evaluate the importance of different components in 3DNEL MSIGP. We evaluate *3DNEL MSIGP (SurfEMB initialization + MCMC)*, which replaces the enumeration-based pose hypotheses generation with SurfEMB predictions and does MCMC finetuning with 3DNEL. This leads to $2.73\%$ improvements over SurfEMB but does not outperform 3DNEL MSIGP, illustrating the effectiveness of MCMC finetuning and the value of having multiple pose hypothe-

| | Scene ID | | | | | | | | | | | |
|---|---|---|---|---|---|---|---|---|---|---|---|---|
| | 48 | 49 | 50 | 51 | 52 | 53 | 54 | 55 | 56 | 57 | 58 | 59 |
| SurfEMB Single Frame | 77.6% | 67.0% | 83.7% | 91.3% | 80.0% | 59.8% | 88.4% | 76.7% | 70.5% | 77.3% | 92.4% | 84.1% |
| 3DNEL MSIGP Single Frame | 71.9% | 77.5% | 83.1% | 87.7% | 87.5% | 84.1% | 88.4% | 80.4% | 82.8% | 85.3% | 94.3% | 86.4% |
| 3DNEL Object Tracking | 73.9% | 78.5% | 84.6% | 90.0% | 91.3% | 92.0% | 90.7% | 83.7% | 91.1% | 87.9% | 95.6% | 64.7% |
| 3DNEL Camera Tracking | **81.5%** | **94.7%** | **97.5%** | **97.0%** | **97.0%** | **97.0%** | **97.2%** | **97.5%** | **96.8%** | **92.2%** | **98.0%** | **97.0%** |

Table 2: **Extending 3DNEL to object and camera tracking improves performance compared to SurfEMB and ablations** We apply 3DNEL to object and camera tracking from video. Our results show that we can leverage temporal information to further improve pose estimation accuracy over the single frame setting, which is impossible for bottom-up pose estimation methods like SurfEMB.

ses. We evaluate *3DNEL MSIGP (Initialization only)*, which uses initialiations from the coarse enumerative procedure as pose estimates. This is substantially worse, demonstrating the importance of MCMC finetuning. We finally evaluate *3DNEL MSIGP (No 2D Detection)*, which uses spherical voting to aggregate evidence from entire query embedding images to generate pose hypotheses. This uses much less information than SurfEMB (no 2D detection and mask prediction), yet results are not far off, demonstrating the effectiveness of our enumeration-based pose hypotheses generation.

In the last section, we contextualize 3DNEL MSIGP's performance in relation to methods that depend on real data. 3DNEL MSIGP outperforms several methods, despite no exposure to real data before test time. We also observe that previous bottom-up approaches that combine RGB and depth information rely heavily on annotated real data. FFB6D, for example, has average recall of $85.5\%$ but this drops to $75.8\%$ without real data. We hypothesize this is because these methods essentially overfit to the characteristics of noisy depth and struggle to generalize to real RGB-D images without having seen them at training. In contrast, as a result of our principled probabilistic modeling of depth information, we can transfer from simulated data to real data without sacrificing performance.

## 4.2 ROBUSTNESS

Figure 4 contains a more detailed comparison between SurfEMB and 3DNEL MSIGP. Across all YCB-V test images, there are 4123 object instances. Each point on the scatter plots corresponds to an object instance, and the point's $x$ and $y$ coordinates are the pose prediction error of 3DNEL MSIGP and SurfEMB, respectively. Points above the dashed line correspond to object instances for which 3DNEL MSIGP had a lower prediction error than SurfEMB. Figure 4(a) shows the scatter plot for all 4123 predictions and Figure 4(c) shows the scatter plots for 6 representative object classes. Figure 4(b) shows, across a range of error thresholds, the number of pose predictions with error above that threshold. 3DNEL MSIGP significantly and consistently reduces the the number of high-error pose predictions, highlighting 3DNEL MSIGP's robustness compared to bottom-up approaches.

We find many instances, especially concentrated within a few object classes, for which SurfEMB is systematically making bad pose predictions. This is primarily a result of the noisy neural embeddings, which SurfEMB's PnP+RANSAC procedure does not handle well. 3DNEL MSIGP is able to correctly estimate the object's pose as a result of enumeration-based pose hypotheses generation, which can robustly handle noisy neural embeddings and identify a variety of pose hypotheses. 3DNEL MSIGP also improves over SurfEMB on test images where the 2D object detector that SurfEMB relies on fails. SurfEMB uses a neural 2D object detector and only makes pose predictions within the predicted bounding box. In contrast, 3DNEL MSIGP considers the full image using spherical voting and therefore can make reasonable pose estimates in spite of missing or wrong 2D detections. Out of the 4123 object instances, there are 174 object instances for which there is no 2D detection and therefore no prediction from SurfEMB. Figure 5 and Figure A4 in the appendix visualize 3DNEL MSIGP and SurfEMB's pose predictions for representative YCB-V test images.

## 4.3 EXTENSION TO OBJECT AND CAMERA TRACKING FROM VIDEO

We demonstrate that, due to its probabilistic generative formulation, we can easily extend 3DNEL to support object and camera tracking from video using principled probabilistic inference in the same model without task specific retraining. We extend the our single-frame 3DNEL to the multi-timestep setup by introducing a simple dynamics prior that samples the object pose at time $t + 1$ uniformly from the space of poses with position at most 3cm away from the object pose at time $t$.

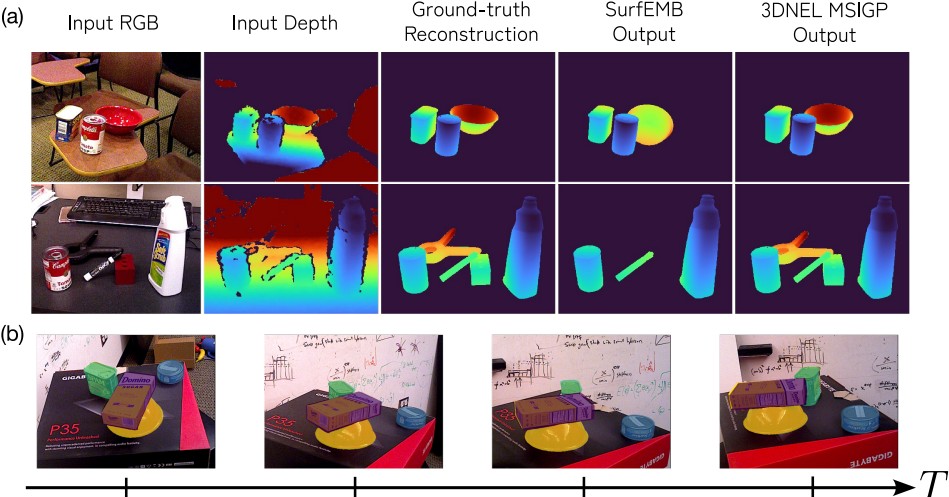

Figure 5: **Qualitative results of 3DNEL used for pose estimation and tracking** (a) Results of 3DNEL MSIGP for 6D pose estimation (b) Results of camera tracking from video using 3DNEL. Predicted object poses are overlayed on each video frame. The $x$-axis indicates time evolution.

For object tracking, we initialize the object poses at the first frame using our single-frame prediction, and apply MCMC finetuning with ICP and random walk proposals for subsequent frames, taking into account the dynamics prior. For camera tracking, we initialize object poses at the first frame to ground truth annotations to avoid introducing systematic errors. We again apply MCMC finetuning with ICP and random walk proposals. However, we further assume we know the scene is static and only the camera moves, which translates into jointly updating the poses of all objects in a scene.

Results in Table 2 show that the same inference procedure can readily handle such extensions, taking into account the dynamics prior for object tracking and the knowledge of a static scene for camera tracking within the same probabilistic generative model. We observe comprehensive improvements over single frame predictions, except for object tracking in scene 59 where we make large errors in the first frame with our single frame predictions and are not able to recover in later frames.

Finally, to highlight the benefits of modeling uncertainty, a key advantage enabled by 3DNEL's probabilistic formulation, we consider a challenging object tracking setup with highly occluded objects. We generate synthetic videos containing two YCB objects, in which one object moves across the scene and becomes fully ocluded by the other static object before reappearing. Methods that maintain only a single object pose estimation for each object tend to lose track of the moving object once it becomes occluded. However, by leveraging Sequential Monte Carlo (Liu & Chen, 1998) to properly model the uncertainty in the pose of the moving object when it is occluded, we are able to achieve reliable tracking in this challenging setup. See Appendix A.7 for details.

## 5 DISCUSSION

This paper presents 3DNEL, a 3D likelihood for RGB-D images based on neural embeddings, and demonstrated SOTA performance in sim-to-real 6D object pose estimation on YCB-V. However, the current work has several limitations. Since 3DNEL is formulated to explain the whole scene, inference with 3DNEL sometimes moves highly occluded objects to explain visible backgrounds. In addition, certain objects are difficult to capture with depth cameras because of their reflective surfaces, which can lead to degraded pose estimation due to the importance of depth cues in 3DNEL. We expect a background model and an outlier model that can properly fall back to use only RGB data when depth is missing will help address these issues. Finally, while 3DNEL improves pose estimation accuracy and robustness, this comes at the cost of increased computation. This is due in part to the lack of performance engineering efforts, and to the non-standard nature of our inference pipeline, which lacks highly optimized software implementations. In future work, we plan to additionally leverage batched GPU rendering and compilation, which will enable a large number of MCMC proposals to be evaluated in parallel and therefore significantly speed up our pipeline.

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

# A  APPENDIX

## A.1  REVIEW OF SURFEMB

Our noise model on RGB information build on SurfEMB (Haugaard & Buch, 2022) which learns neural embeddings to establish dense 2D-3D correspondences via contrastive learning. For each object class $t \in \{1, \cdots, M\}$, SurfEMB learns a pair of neural networks: the *query model* $f_t : \{0, \cdots, 255\}^{H \times W \times 3} \mapsto \mathbb{R}^{H \times W \times E}$ and the *key model* $g_t : \mathbb{R}^3 \mapsto \mathbb{R}^E$ . The *query model* transforms an observed RGB image $\mathbf{I}$ into query embeddings $\mathbf{Q}^t$, while the *key model* transforms a rendered object coordinate image $\tilde{\mathbf{X}}$ into a set of key embeddings.

For a given 2D pixel location on the observed image with query embedding $\mathbf{q}$, SurfEMB specifies a surface correspondence distribution $\mathbb{P}_{RGB}(\tilde{x}|\mathbf{q}, t) \propto \exp(\mathbf{q}^T g_t(\tilde{x}))$ for each object class $t$. To normalize this surface correspondence distribution, for each object class $t$, we subsample uniformly across the object's surface to get a set $Z_t$ of surface 3D coordinates in object frame. Then, given a pixel with query embedding $\mathbf{q}$, we calculate the probability that this pixel corresponds to a surface point $\tilde{\mathbf{x}} \in Z_t$ on object class $t$ as:

$$\mathbb{P}_{RGB}(\tilde{\mathbf{x}}|\mathbf{q}, Z_t, t) = \frac{\exp(\mathbf{q}^T g_t(\tilde{\mathbf{x}}))}{\sum_{\mathbf{x} \in Z_t} \exp(\mathbf{q}^T g_t(\mathbf{x}))} \tag{2}$$

## A.2  MORE DETAILS ON THE ENERGY-BASED FORMULATION OF 3DNEL

Since we are working with an energy-based formulation (Equation 1), to make the probability distribution properly defined we need to make sure the normalization constant, i.e. the sum of the energy function over all $\mathbf{I}$ and $\mathbf{C}$

$$\sum_{\mathbf{I}} \int_{\mathbf{C}} \prod_{\mathbf{c}} \left( \epsilon \mathbb{P}_{background}(\mathbf{c}; B) + \frac{1-\epsilon}{\tilde{K}} \sum_{\tilde{\mathbf{c}}:\tilde{s}>0} \mathbb{P}_{depth}(\mathbf{c}|\tilde{\mathbf{c}}; r) \mathbb{P}_{RGB}(\tilde{\mathbf{x}}|\mathbf{q}^{\tilde{s}}, \tilde{s}) \right)$$

is finite and well-defined.

For RGB images of size $H \times W$, since each pixel has only 256 values, there are at most $256^{H \times W \times 3}$ RGB images of size $H \times W$ which is a finite number. Since the value of the energy function is less than 1 for any given $\mathbf{I}$ and $\mathbf{C}$, summing over a finite number of $\mathbf{I}$ and integrating over a bounded region for $\mathbf{C}$ gives us a finite normalization constant, making the probability distribution well-defined.

## A.3  DETAILS ON ENUMERATION-BASED POSE HYPOTHESES GENERATION

In Algorithm 1 we present a detailed description of our enumeration-based pose hypotheses generation process that is introduced in Section 3.3. In practice we also optionally apply non-max suppression in the `TopPositions` function in the algorithm to make sure the promising positions we identify are well spread out to cover different parts of the image and better represent uncertainty.

## A.4  DETAILS ON MCMC FINETUNING

Our pose hypotheses proposals go through the $N$ objects present in a scene multiple times. For object $i$ of class $t_i$, the pose hypotheses proposals sequentially go through all pose hypotheses for class $t_i$. For each one of these pose hypotheses, the pose hypotheses proposals sample a pose centered at the given pose hypothesis, and propose to change the pose of object $i$ to the sampled pose. Going through the objects more times help improve performance, at the cost of increased computation. In practice we find going through all objects twice strikes a good balance between performance and speed.

## A.5  DETAILS ON INFERENCE PIPELINE IMPLEMENTATION

### A.5.1  DISCRETIZING THE ROTATION SPACE

---

**Algorithm 1:** Enumeration-based Pose Hypotheses Generation

---

```
/* Basic setups                                                    */
```
**Object-specific:** A set of surface points $Z_t$ for object class $t$, query model $f_t$, key model $g_t$, $n_k$
    keypoints with object frame coordinates $x_1^*, \cdots, x_{n_k}^* \in \mathbb{R}^3$.

**Spatial discretization:** Camera frame coordinates of the center of the boundary voxel $y \in \mathbb{R}^3$,
    size of the voxel grid $(L_x, L_y, L_z)$, diameter of the voxels $d > 0$.

**Orientation discretization:** $n_r$ representative orientations $\mathbf{R}_1, \cdots, \mathbf{R}_{n_r} \in \mathbb{SO}(3)$.

**Parameters:** Number of pose hypotheses to generate $n_p$, number of top positions $n_t$.

**1**

---

```
/* Enumeration-based pose hypotheses generation                    */
```
**Input:** RGB image $\mathbf{I} \in \mathbb{R}^{H \times W \times 3}$, observed point cloud $\mathbf{C} \in \mathbb{R}^{H \times W \times 3}$.
**Output:** Top scoring pose hypotheses $\mathbf{P}_1^t, \cdots, \mathbf{P}_{n_p}^t \in \mathbb{SE}(3)$.

**2** $\mathbf{Q} \leftarrow f_t(\mathbf{I})$; `// Get query embeddings` $\mathbf{Q} \in \mathbb{R}^{H \times W \times E}$ `from the RGB image` $\mathbf{I}$
**3** $\mathbf{V}^0 \leftarrow \text{Voting}(\mathbf{Q}, \mathbf{C}, (0,0,0))$;   `// Aggregation for object center` $(0,0,0)$
**4** **for** $i \leftarrow 1$ **to** $n_k$ **do**                    `// Aggregation for` $n_k$ `keypoints.`
**5** $\quad\lfloor \mathbf{V}^i \leftarrow \text{Voting}(\mathbf{Q}, \mathbf{C}, x_i^*)$;                    `//` $\mathbf{V}^i \in \mathbb{R}^{L_x \times L_y \times L_z}$

```
/* Identify top positions based on V⁰'s largest entires.    */
```
**6** $l_1, \cdots, l_{n_t} \leftarrow \text{TopPositions}(\mathbf{V}^0)$;                    `//` $l_1, \cdots, l_{n_t} \in \mathbb{R}^3$
**7** **for** $i \leftarrow 1$ **to** $n_t$, $j \leftarrow 1$ **to** $n_r$ **do**
**8** $\quad\lfloor s_{i,j} \leftarrow \text{Scoring}(l_i, \mathbf{R}_j, \mathbf{V}^1, \cdots, \mathbf{V}^{n_k})$;          `// Heuristic pose scoring`
**9** $\mathbf{P}_1^t, \cdots, \mathbf{P}_{n_p}^t \leftarrow \text{RankByScore}(l_1, \cdots, l_{n_t}, s_{i,j}, i = 1, \cdots, n_t, j = 1, \cdots, n_r)$;
**10** **return** $\mathbf{P}_1^t, \cdots, \mathbf{P}_{n_p}^t$;

**11**

---

```
/* Voting and heuristic scoring                                    */
```
**12** **def** $\text{Voting}(\mathbf{Q}, \mathbf{C}, x^*)$              `// Voting-based evidence aggregation`
**13** $\quad \mathbf{V} \leftarrow \mathbf{0}$;              `// Initialize` $V \in \mathbb{R}^{L_x \times L_y \times L_z}$ `to all 0 array`
**14** $\quad$ **for** $i \leftarrow 1$ **to** $H$, $j \leftarrow 1$ **to** $W$ **do**
**15** $\quad\quad x \leftarrow \arg\max_{\tilde{x}} \mathbb{P}_{RGB}(\tilde{x}|Q_{i,j}, Z_t, t), p_{i,j} \leftarrow \max_{\tilde{x}} \mathbb{P}_{RGB}(\tilde{x}|Q_{i,j}, Z_t, t)$;
**16** $\quad\quad$ **for** $u \leftarrow 1$ **to** $L_x$, $v \leftarrow 1$ **to** $L_y$, $w \leftarrow 1$ **to** $L_z$ **do**
**17** $\quad\quad\quad c \leftarrow (y_1 + (u-1)d, y_2 + (v-1)d, y_3 + (w-1)d)$;
**18** $\quad\quad\quad$ **if** $||\mathbf{C}_{i,j} - c||_2 \approx ||x - x^*||_2$ **then**
**19** $\quad\quad\quad\quad \lfloor \mathbf{V}_{u,v,w} = \mathbf{V}_{u,v,w} + p_{i,j}$
**20** $\quad$ **return** $\mathbf{V}$
**21** **def** $\text{Scoring}(l, \mathbf{R}, \mathbf{V}^1, \cdots, \mathbf{V}^{n_k})$              `// Heuristic pose scoring`
**22** $\quad s \leftarrow 0$;              `// Initialize score to 0`
**23** $\quad$ **for** $i \leftarrow 1$ **to** $n_k$ **do**
**24** $\quad\quad x \leftarrow \mathbf{R}x_i^* + l$;    `// Location of` $x_i^*$ `in world frame for pose` $l, \mathbf{R}$
**25** $\quad\quad (u, v, w) \leftarrow Round[(x - y)/d]$; `// Identify corresponding voxel of` $x$
**26** $\quad\quad s = s + \mathbf{V}_{u,v,w}^i$;              `// Heuristic scoring`
**27** $\quad$ **return** $s$

---

We discretize $\mathbb{SO}(3)$ into 6400 representative orientations. We generate these orientations by first picking 200 points roughly uniformly on the unit sphere using the Fibonacci sphere. The 6400 representative orientations are genearted by first rotating the axis $(0, 0, 1.0)$ to point to one of the 200 points, followed by one of 32 in-plane rotations around the axis.

### A.5.2  USING ADDITIONAL 2D DETECTION AND MASK PREDCITION FROM SURFEMB

In the best performing setup, we leverage the same 2D detector used in SurfEMB as part of the pose hypotheses generation process. Although our spherical voting procedure can robustly aggregate information from the entire image to generate pose hypotheses, as demonstrated by the competitive performance of 3DNEL MSIGP (No 2D detection) (Abalations in Table 1), in practice the query embedding images for many objects are very noisy and tend to hurt performance. Empirically,

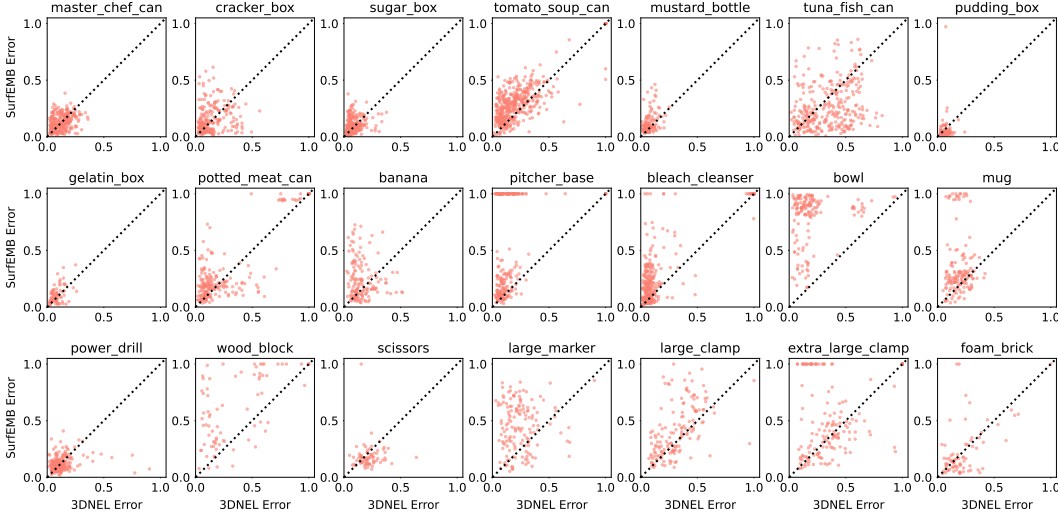

Figure A1: We compare the prediction error (using the VSD error metric) of SurfEMB and 3DNEL MSIGP across all 4123 object instances in the YCB-V test dataset, with each instance represented as a point on this scatter plot. In the main text Figure 4(a), we show the scatter plot across all objects. Here, we show the results per object.

we observe that by additionally using 2D detections, we can focus the spherical voting process on regions of the observed image that is likely relevant for the objects, and further improve performance.

For each object class in the scene, there can be multiple 2D detections. For each 2D detection, we do spherical voting just within the detector crop and generate 80 pose hypotheses per detector crop. When there is a missing 2D detection, we obtain the query embeddings by upsampling the input RGB image by 1.5x, and do spherical voting on the whole image. In such cases, when we identify top positions, we additionally do non-max suppression with a filter size of 10 to spread the top-scoring positions out. For each such top-scoring position we identify top 2 orientation, and we consider all top-scoring positions and generate in total 30 pose hypotheses.

SurfEMB's query embedding model was trained on image crops from 2D detections rescaled to $224 \times 224$, but can be applied to RGB images of any size. Empirically we observe that due to Sur-fEMB's rescaling of the 2D detection crops to $224 \times 224$, the query embeddings for some objects are only informative when we properly rescale the input RGB image. To account for this, we estimate the best scale by scoring the top pose hypotheses (using our heuristic scoring as described in Section 3.3) from each 2D detector crop and pick the scale of the highest-scoring 2D detector crop. Concretely, when there are multiple 2D detections for an object class, we score the corresponding scales with the mean heuristic score of the top 5 pose hypotheses from that detector crop, and use the highest scoring scale to obtain the query embeddings for the object class. When there is no 2D detections we fall back to a default scale of $1.5$.

### A.5.3 IMPLEMENTATION DETAILS AND HYPERPARAMETERS

We use OpenGL for rendering, and use Taichi (Hu et al., 2019) for both spherical voting and 3DNEL evaluation. A Python implementation reproducing the results reported in the paper will be released upon acceptance.

As we describe in the main text, we pick hyperparameters by visually inspecting detection results on a small number of real training images that are outside the test set.

We select $n_k = 8$ keypoints from the surface of each object class, and use $y = (-350.0, -210.0, 530.0), L_x = 129, L_y = 87, L_z = 168$ and $d = 5.0$ to make the voxel grids large enough to cover all the keypoints that can be present in the camera frame. Here the units for the values in $y$ and $d$ are mm.

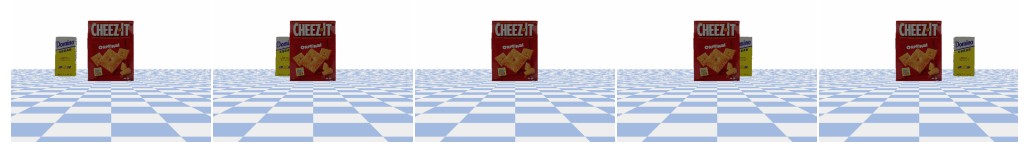

Figure A2: Visualizing an example synthetic video we use to demonstrate the benefits of modeling uncertainty. We render a short video containing two YCB objects, the cracker box and the sugar box. The cracker box is static, while the sugar box is moving and gets completely occluded by the cracker box before reappearing. Tracking is challenging in such setups for methods that only use a single object pose estimation. However by properly modeling uncertainty using the posterior distribution on object poses from 3DNEL through Sequential Monte Carlo, we are able to achieve successful tracking. The $x$-axis represents time evolution.

We use $r = 5.0$ in evaluating 3DNEl for all our experiments. We implement 3DNEL with Taichi to leverage powerful GPUs. When identifying points in the rendered point cloud $\tilde{\mathbf{C}}$ (organized as an $H \times W \times 3$ image) that is within distance $r$ from a point $(i, j)$ in the observed point cloud $\mathbf{C}$, to further speed up 3DNEL evaluation, we only look at the points from the patch of size $(10, 10)$ centered at $(i, j)$.

For MCMC finetuning, we go through all pose hypotheses twice in the pose hypotheses proposals. We go through all objects 5 times in the ICP proposals. We go through all objects 50 times in the random walk proposals. Each time we go through an object in the random walk proposals, we propose 7 types of random pose perturbations.

## A.6 ADDITIONAL ROBUSTNESS RESULTS

In Figure A1 we include additional robustness results.

## A.7 BENEFITS OF MODELING UNCERTAINTY: AN ADDITIONAL TRACKING EXPERIMENT

To highlight the benefits of modeling uncertainty, a key advantage enabled by 3DNEL's probabilistic formulation, we present an additional experiment on object tracking in a synthetic video containing two YCB objects, the cracker box and the sugar box. The cracker box remains static while the sugar box moves across the scene. The sugar box gets fully occluded by the cracker box before reappearing. We generated this video in PyBullet (Coumans & Bai, 2016–2021) using the textured mesh models from YCB-V dataset for the cracker box and sugar box. Some example frames from the synthetic video are shown in Figure A2.

This is a challenging setup for methods that do not model uncertainty but only rely on a single estimation of the object pose, as such methods tend to lose track once the moving object becomes severely occluded. We find that when 3DNEL is used to only produce a single estimation of the object pose, tracking indeed fails. However, when we apply 3DNEL for tracking using Sequential Monte Carlo, we are able to approximate the posterior distribution over the object's pose at each timestep and capture the uncertainty in the pose of the moving object when it gets occluded (rather than a single estimation of the object pose). We find that this can help us accurately trace the object and regain track of the moving object when it reappears from occlusion.

For Sequential Monte Carlo, we work in a particle filtering setup, and assume a simple dynamics model in which the object's translation is sampled from a multivariate Gaussian centered at the translation from the previous timestep, and the orientation is sampled from a Von-Mises-Fisher distribution centered at the orientation from the previous timestep.

In Figure A3, we visualize the tracking results and the inferred posterior distribution (approximated via particles in a particle filtering setup) at different timesteps. When the sugar box becomes occluded, we see the posterior distribution on the pose of the sugar box spread out across the many positions the sugar box could be at behind the cracker box. When the sugar box eventually reappears

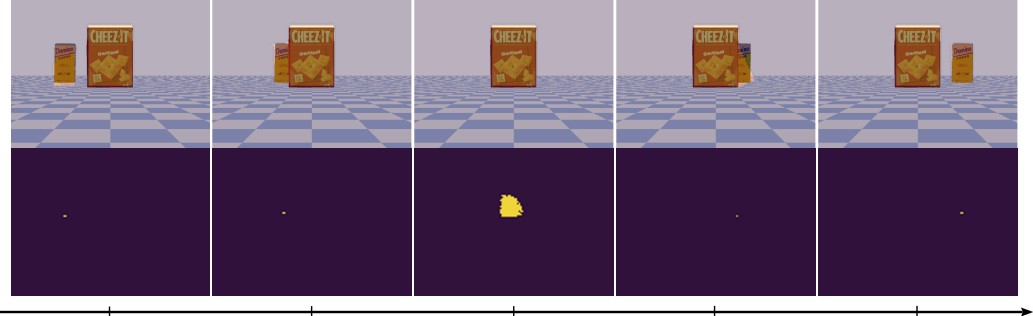

Figure A3: Visualizing object tracking in the challenging setup with highly occluded objects. We use a particle filtering setup. Top row visualizes overlays of the predicted object poses and the RGB images. Object poses are derived from the particle that has the highest posterior probability. Bottom row visualizes how the posterior distribution on the pose of the sugar box (represented using the particles) evolves as the sugar box moves across the scene. As we can see, once the sugar box gets occluded, the posterior distribution spreads out across the scene to model the uncertainty we have on the pose of the sugar box. When the sugar box reappears, the posterior distribution becomes concentrated again and the system is able to regain track of the sugar box. The $x$-axis indicates time evolution.

we can regain track of its pose. On the other hand, methods that maintain only a single object pose estimation will almost surely lose track once the moving object gets occluded.

Note that for this experiment, we leverage an improved pure JAX implementation of the 3DNEL likelihood. We implemented a JAX depth image renderer, which enables a large number of 3DNEL evaluations in parallel. This allows us to achieve real-time tracking performance for $480 \times 640$ images on a machine with a single NVIDIA A100 GPU. Of course the involved data are synthetic and much simpler. However this is a promising early step towards significantly speeding up the 3DNEL inference process.

## A.8 ADDITIONAL VISUALIZATIONS OF SURFEMB AND 3DNEL MSIGP PREDICTIONS ON YCB-V TEST IMAGES

In Figure A4 we include additional visualizations of SurfEMB and 3DNEL MSIGP predictions on YCB-V test images.

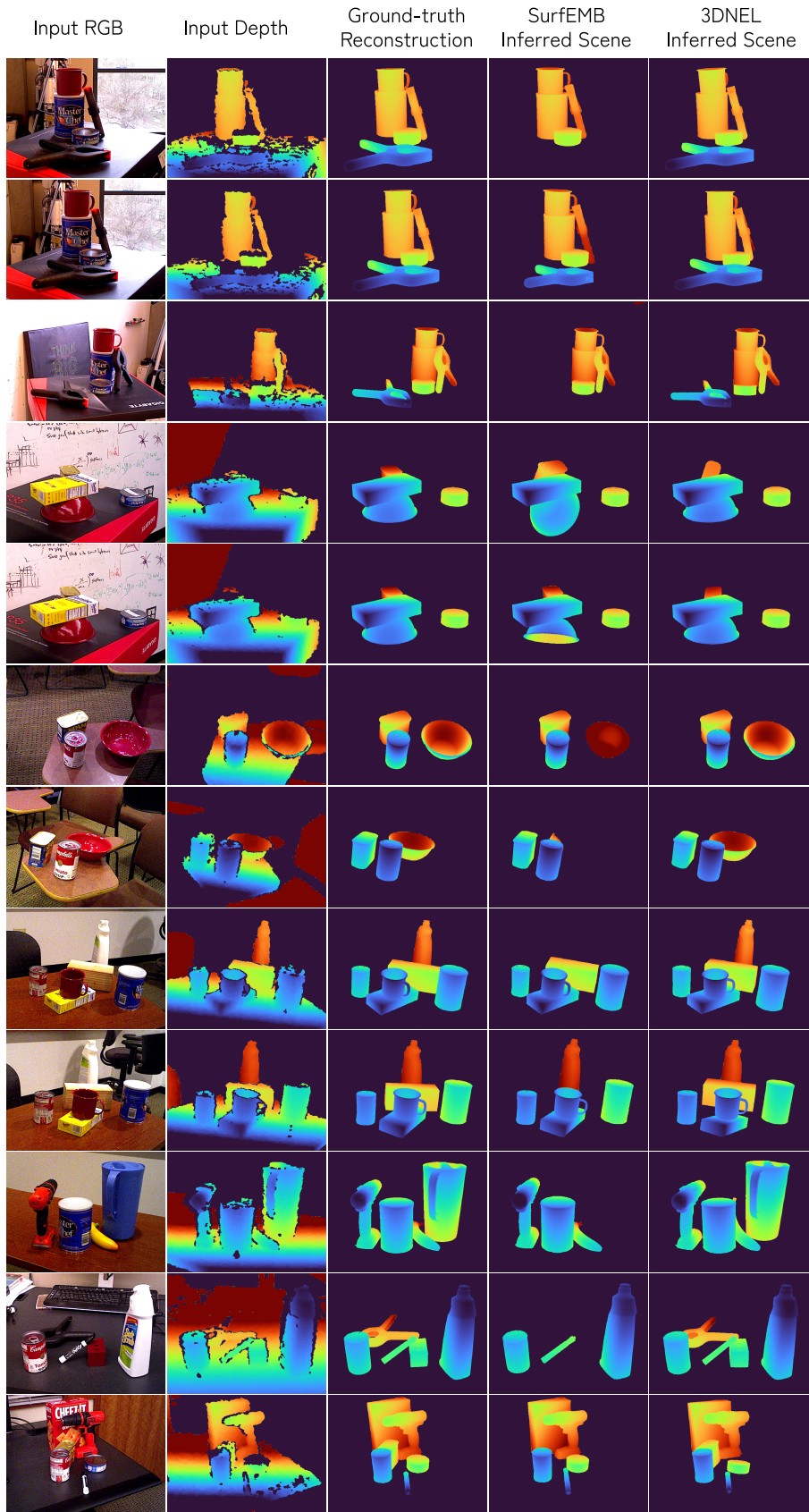

Figure A4: Additional visualizations of SurfEMB and 3DNEL MSIGP predictions on YCB-V test images

