# OpenReview forum: "3D Neural Embedding Likelihood for Robust Sim-to-Real Transfer in Inverse Graphics"
_ICLR.cc/2023/Conference — Submitted to ICLR 2023_

### Official Review · Reviewer_a254 · 2022-10-24

**Confidence:** 3
**Clarity, Quality, Novelty And Reproducibility:** The idea is clear and easy to follow.
**Correctness:** 3
**Technical Novelty And Significance:** 3
**Empirical Novelty And Significance:** 2
**Recommendation:** 6

**Strength And Weaknesses:**

+ A 2d/3d correspondence approach to handle sim-to-real pose estimation.
+ the probabilistic formulation in Eqn (2) and (3) makes sense.
+ Spherical voting idea is interesting, related in spirit to hough voting.
+ Experimentally, the model achieves strong performance compared to SOTA Ransac-based-proposal approaches.

**Summary Of The Paper:**

The author proposed an inverse-graphics approach for robust object pose estimation: the model first evaluates pose performance on discretized pose hypothesis with a spherical voting, then finetunes the pose with MCMC. The performance on YCB-dataset is strong and achieves the state-of-the-art.

**Summary Of The Review:**

In summary, the author proposed an interesting idea of 2D/3D-correspondence-based  approach to solve the object pose estimation problem. The idea is clearly presented and easy to follow. And the experimental results on YCB is strong.

I have a few questions:
* Any insight why the proposed approach can be better than RANSAC+SurfEMB? the evidence model proposed in Eqn (1) could be covered by RANSAC if a correct correspondence is proposed to generate a hypothesis.
* Robustness: if a RANSAC-based algorithm tries enough hypothesis, will it be more robust as well?

---

> ### Author Response · Authors · 2022-11-13
> **Thank you for your constructive feedback!**
>
> Thank you for your comments and suggestions. We address specific questions below.
>
> > Any insight why the proposed approach can be better than RANSAC+SurfEMB? the evidence model proposed in Eqn (1) could be covered by RANSAC if a correct correspondence is proposed to generate a hypothesis.
>
> Compared with PnP+RANSAC+SurfEMB, our method is more principled and the improved performance is due to the following key advantages:
>
> - Our spherical voting process is entirely feedforward, is fast and efficient, makes use of both RGB (in the form of the correspondence distributions) and depth information, and can robustly aggregate information from the entire image. This is in contrast with the PnP+RANSAC which requires lots of iterations, uses only RGB information, and heavily relies on the initial 2D detections for object localization.
> - 3DNEL combines RGB and depth information into a unified probabilistic model, while SurfEMB does scoring only with RGB information and uses depth information in a separate, heuristic refinement stage.
>
> We clarify that Eqn (1) in the original manuscript is from SurfEMB, and is the exact correspondence distribution SurfEMB uses (together with PnP) to generate pose hypotheses. However, Eqn (1) can only use RGB information, and depends heavily on the initial 2D detections for localizing the objects. In contrast, our spherical voting procedure can make use of both RGB and depth information and robustly aggregate information from the entire image to generate pose hypotheses, and 3DNEL also evaluates the pose hypotheses using a principled combination of RGB and depth information.
>
> > Robustness: if a RANSAC-based algorithm tries enough hypothesis, will it be more robust as well?
>
> It is unlikely that a RANSAC-based algorithm would achieve the same robustness as 3DNEL MSIGP even with a large number of pose hypotheses. To mention two issues:
>
> 1. One issue concerns the difficulty of generating high-quality pose hypotheses just from RGB information. An example is the red bowl object from the YCB-V dataset (e.g. Figure 5(a) top row in the updated manuscript), where SurfEMB struggles a lot since PnP+RANSAC uses only RGB information and generates low-quality pose hypotheses. In contrast 3DNEL combines RGB and depth information into a unified probabilistic model and can robustly handle such cases.
> 2. Another issue concerns missing 2D detections. Since PnP+RANSAC heavily relies on the initial 2D detections for localizing the objects, when there are missing 2D detections (e.g. Figure 5(a) bottom row in the updated manuscript), PnP+RANSAC cannot even generate pose hypotheses. In contrast, our proposed spherical voting procedure can robustly aggregate information from the entire image and can handle such cases without issues.
>
> We thank you again for the very detailed and constructive comments. Please don’t hesitate to let us know of any additional comments on the paper. We are very happy to address them.

---

### Official Review · Reviewer_4Xvb · 2022-10-24

**Confidence:** 4
**Correctness:** 2
**Technical Novelty And Significance:** 2
**Empirical Novelty And Significance:** 2
**Recommendation:** 6

**Clarity, Quality, Novelty And Reproducibility:**

While the paper is mostly easy to follow, the technical description of the method appears quite convoluted and difficult to understand. Several aspects of the method need to be described more clearly. To provide a few examples:

- What is meant by latent 3D scene?
- Section 3.2 refers to embeddings Q^t, while ' the key model an input 3D model coordinate into an embedding'. What is the difference between these embeddings? As is this is not clear.
- In Section 3.3. what is meant by 'the learned neural embedding share used to define a likelihood that computes the probability of an RGB-D image given a hypothesized latent 3D scene'? Why hypothesized latent 3D scene?
- The motivation Section 3.3 is difficult to follow and should be described more clearly. Why are the semantic segmentation maps observed? How exactly is the object coordinate array defined?
- How exactly is the 'latent point cloud generated'? Why is it latent?
- What is meant by 'additionally have a uniform distribution on the bounded region as an outlier component'? What is meant by 'outlier component'?
- Why is the 'normalization constant' well defined? Why is there a finite number of RGB images?
- Table 1: Why are the results only reported for 5 runs? Wouldn't it make sense to validate on a larger number?
- Figure 4: What is the arrow on the x-axis supposed to indicate?
- I would suggest to provide runtime/inference runtime performance measures. This would especially be interesting for the MCMC + ICP experiment discussed in Section 5.3.

As is, not enough information is provided to allow for reproducing the proposed method.

**Details Of Ethics Concerns:**

I do not have any concerns towards ethical considerations.

**Strength And Weaknesses:**

Strengths:

- The topic of using synthetic data for effectively training neural networks is interesting and important.
- The proposed approach of combining RGB and depth images by employing neural embeddings seems novel although somewhat marginal compared to SurfEMB.
- The presented experiments are mostly sounds and the results seem promising.

Weaknesses:

- The technical section of the paper is quite convoluted and difficult to follow.
- Contrary to the author's claims it does not seem as if the method is easy to apply to other tasks and training setups.
- The method is only shown to work on pose estimation.

**Summary Of The Paper:**

The presented paper aims to introduce a method for the robust sim-to-real transfer to address the domain gap between synthetically generated data and real data. The goal of the method is to support training a model with synthetic data so as to generalize to real data. The proposed method -- called 3DNEL -- uses neural embeddings to predict 2D-3D correspondences from RGB and depth images. 3DNEL is used as part of a multi-stage inverse graphics pipeline that is tested on a 6D pose prediction task.


**Summary Of The Review:**

Overall, I have the impression that the topic this paper address is important and interesting to the ICLR community. However, it its current form I cannot recommend accepting this work. This is due to the following reasons:

- The technical description of the method appears quite convoluted and is difficult to follow. The method section needs to be rewritten carefully and more details need to be provided.
- While the method seems to be sound for the setup defined in the paper, it is not clear how it could be used for other tasks or setups.
- Compared to SurfEMB the technical contribution appears somewhat marginal.

---

> ### Author Response · Authors · 2022-11-13
> **Thank you for your constructive feedback (part 1)**
>
> Thank you for your comments and suggestions. As suggested, we have carefully rewritten the two methods sections (Sections 3 and 4 in the original manuscript), with the changes highlighted in blue in the updated manuscript. We also appreciate your constructive comments, and address your questions point-by-point as below.
>
> > What is meant by latent 3D scene?
>
> In the updated manuscript, we have changed all “latent 3D scene” to just “3D scene description” to improve clarity. In the newly added Preliminaries section (Section 3.1 in the updated manuscript), we make it clear that we aim to build a likelihood $\mathbb{P}(\text{Observed RGB-D image} | \text{3D scene description})$, and that our 3D scene description is given by the number of objects in a scene $N$, the classes of the objects $t_1, \cdots, t_N$, and the poses of the objects $P_1, \cdots, P_N$.
>
> > Section 3.2 refers to embeddings Q^t, while ' the key model an input 3D model coordinate into an embedding'. What is the difference between these embeddings? As is this is not clear.
>
> The query embeddings are derived from the observed RGB images, while the key embeddings are derived from the rendered object coordinate images. Both are $E$-dimensional vectors, and we use their inner products to calculate the dense 2D-3D correspondence distributions.
>
> We have carefully rewritten the paragraph reviewing SurfEMB in the main text (Section 3.1 in the updated manuscript) to make it more self-contained and clarify the difference between the two kinds of embeddings, and an additional detailed review of SurfEMB in Appendix A.1.
>
> > In Section 3.3. what is meant by 'the learned neural embedding share used to define a likelihood that computes the probability of an RGB-D image given a hypothesized latent 3D scene'? Why hypothesized latent 3D scene?
>
> We have made a few changes to make this statement clear. We changed the “hypothesized latent 3D scene” to “3D scene description”. We clarified that 3DNEL is a likelihood that calculates $\mathbb{P}(\text{Observed RGB-D image} |  \text{3D scene description})$. We described in detail how we process both the 3D scene description and the observed RGB-D image to obtain point cloud images and the neural embeddings for likelihood evaluation, and how we use the processed 3D scene description and observed RGB-D images to evaluate 3DNEL (Section 3.2 in the updated manuscript).
>
> > The motivation Section 3.3 is difficult to follow and should be described more clearly. Why are the semantic segmentation maps observed? How exactly is the object coordinate array defined?
>
> The semantic segmentation maps and the object coordinate array are just used to illustrate the idea of modeling 2D-3D correspondence, but are not actually observed.
>
> We now see that the original motivation can be confusing. To remedy this, we have carefully rewritten the 3DNEL section, and directly describe the energy-based formulation of 3DNEL using processed 3D scene description and observed RGB-D images, and also clearly describe how we process the 3D scene description and observed RGB-D image for likelihood evaluation (Section 3.2 in the updated manuscript).
>
> > How exactly is the 'latent point cloud generated'? Why is it latent?
>
> The latent point cloud is generated by first rendering the 3D scene description to a rendered depth image, and then unprojecting using camera intrinsics to get the point cloud.
>
> Seeing that the term “latent” is confusing, in our updated manuscript we have changed all the use of “latent” to just “rendered”.
>
> > What is meant by 'additionally have a uniform distribution on the bounded region as an outlier component'? What is meant by 'outlier component'?
>
> In the observed image, in addition to pixels that are associated with objects, there are also background pixels. We use the uniform distribution on the bounded region as a component in our mixture distribution formulation to model background pixels in the observed image.
>
> Seeing that the term “outlier component” is confusing, we have switched to just saying this is a component for modeling background, and additionally introduced ${P}_{background}(\mathbf{c}; B) = 1/B$ to make the equation self-explanatory and improve clarity.

---

> ### Author Response · Authors · 2022-11-13
> **Thank you for your constructive feedback (part 2)**
>
> > Why is the 'normalization constant' well defined? Why is there a finite number of RGB images?
>
> This is a technical detail. Seeing that this is confusing in the main text, we have instead moved this sentence to the appendix, and expanded the explanation in the appendix. See Appendix A.2 in the updated manuscript.
>
> To also include the explanation here: since we are working with an energy-based formulation, to make the probability distribution properly defined we need to make sure the normalization constant, i.e. the sum of the energy function over all $I$ and $C$, is finite and well-defined. For RGB images of size $H\times W$, since each pixel has only 256 values, there are at most $256^{H\times W\times 3}$ RGB images which is a finite number. Since the value of the energy function is less than 1 for any given $I$ and $C$, summing over a finite number of $I$ and integrating over a bounded region for $C$ gives us a finite normalization constant, making the probability distribution well-defined.
>
> > Table 1: Why are the results only reported for 5 runs? Wouldn't it make sense to validate on a larger number?
>
> Empirically we observe that the performances across different runs are essentially the same, as reflected by the very small standard deviation in our accuracy numbers. As a result we do not feel it is necessary to validate on a larger number of runs (the performance would likely still be the same).
>
> > Figure 4: What is the arrow on the x-axis supposed to indicate?
>
> The arrow indicates the evolution of time, which is used to make it clear that we are working in a tracking setup.
>
> To make this clear, in the updated manuscript we have added a $T$ on the x-axis and also clearly state in the figure caption that the arrow indicates evolution of time.
>
> > I would suggest to provide runtime/inference runtime performance measures. This would especially be interesting for the MCMC + ICP experiment discussed in Section 5.3.
>
> With the current implementation our inference takes 20-30 seconds per image for pose estimation from static images, and around 15-20 seconds per frame for the tracking experiments. Timings are obtained on a machine with a single A100 GPU. For comparison, the baseline SurfEMB reported a timing of around 5 seconds per image for pose estimation from static images, but cannot be applied to the tracking setup. We have updated the manuscript to include these numbers (see Section 4 in the updated manuscript).
>
> While our current implementation is slower than baselines, we expect significant speedups just with additional performance engineering. To illustrate this, we include an additional experiment in a tracking setup where we implement approximate rendering and parallel likelihood evaluation using JAX and achieve real-time tracking performance, although on simpler and synthetic data. See the experiment description in the general response for details.
>
> > As is, not enough information is provided to allow for reproducing the proposed method.
>
> We have carefully rewritten the methods section, and significantly expanded the appendix, to improve clarity and reproducibility. In addition, as we have mentioned in the manuscript, upon acceptance we will open source a Python codebase reproducing all the results reported in the paper.
>
> > The technical section of the paper is quite convoluted and difficult to follow. The method section needs to be rewritten carefully and more details need to be provided.
>
> As requested by the reviewers, we have carefully rewritten the two methods sections (Section 3 and 4), and significantly expanded the appendix with more details on different aspects of the pipeline. Please also refer to the above responses for all the raised questions.

---

> ### Author Response · Authors · 2022-11-13
> **Thank you for your constructive feedback (part 3)**
>
> > it does not seem as if the method is easy to apply to other tasks and training setups.
>
> We emphasize that our experiments have already demonstrated 3DNEL can be easily extended to both object and camera tracking from video, by additionally taking into account the temporal information that is present in video, all with minimal changes to the model and inference process. We want to point out that this is impossible for SurfEMB, as SurfEMB is a bottom-up pipeline trained specifically for the task of pose estimation and does not have a mechanism to leverage the additional temporal information.
>
> In addition, although in this paper we use object poses to specify our 3D scene description and assume known object mesh models, in general there are many other possibilities, and many do not involve pose estimation at all. One example is to additionally parametrize the object shapes and use inference on multi-object videos to recover separate object 3D models (which now becomes part of the 3D scene description), all using the same 3DNEL and 3D inverse graphics formulation. We are planning to explore this setup in future works.
>
> We added a comment to the caption of Table 2 in the updated manuscript to emphasize our ability to additionally leverage temporal information, and point out that this is impossible for other baseline pose estimation methods like SurfEMB.
>
> > Compared to SurfEMB the technical contribution appears somewhat marginal.
>
> The focus of our work is **not** to propose new learning methods for visual correspondence. We deliberately chose to use the learned neural embeddings from SurfEMB to make it easy to compare with existing baselines. Even with the same underlying model, we observe significant performance improvements. We see obvious ways to further improve performance by tailoring the correspondence learning to the 3DNEL formulation instead reusing pretrained SurfEMB models, but leave these explorations to future work.
>
> We thank you again for the very detailed and constructive comments. Please don’t hesitate to let us know of any additional comments on the paper. We are very happy to address them.

---

### Official Review · Reviewer_aQXb · 2022-10-25

**Confidence:** 3
**Correctness:** 4
**Technical Novelty And Significance:** 3
**Empirical Novelty And Significance:** 3
**Recommendation:** 6

**Clarity, Quality, Novelty And Reproducibility:**

The joined probabilistic formulation for RGB and depth is novel and interesting.
While the basis is a mix of 3dp3 and SurfEMB, bringing them together is a novel idea and the proposal generation seems also different to existing ideas.

**Strength And Weaknesses:**

Strength:

The probabilistic formulation is simple and seems robust.
Proposal generation method based on spherical voting from depth and correspondences is independent of 2d detections priors.
The authors show the usefulness of the probabilistic model to generate possible poses under occlusion.

Weaknesses:

A remaining question is which hypothesis are actually sampled in the MCMC optimization. An ablation for the impact of the different methods Voting, ICP and Random Walk would help to evaluate their impact on the final result.

It seems that at the moment the proposed method leverages the knowledge of how many objects are present in the scene to create the initial proposals, which is a hugh advantage over other methods that do not have this assumption. While it seems like the assumption could be relaxed, it is unclear if the method remains stable in this case.

In 3.3 the authors write "ICP proposals iterate over all objects and use ICP to align each object’s 3D model to the observed 3D point cloud with the point cloud from all the other objects masked out,", however it is unclear how they can mask out all other objects if they only have semantic masks and not instance masks. While using synthetic data learning instance masks would probably not pose a larger problem than predicting SufEmbeddings, it is unclear how these masks are generated at the moment.


Corrections:
In 3.1 the dimensions of Q should probably be HxWxE instead of HxWx3

**Summary Of The Paper:**

This paper proposes a probabilistic generative method leveraging learned 2d-to-3d mappings from SurfEMB and depth (3DP3) that generates and optimizes (MCMC) 6d pose hypotheses based on RGB-D images. The method is purely trained on synthetic images and evaluated on real data showing significant improvements to the baseline method by evaluating multiple hypothesis and refining them using MCMC.

**Summary Of The Review:**

After the rebuttal of the authors the explanation of the method is very clear and the probabilistic formulation shows advantages over existing methods especially also for estimating the uncertainties. There remain a few open points that would be interesting to follow up on, but the proposed idea and results are already showing the benefits of the proposed approach.

---

> ### Author Response · Authors · 2022-11-13
> **Thank you for your constructive feedback (part 1)**
>
> Thank you for your comments and suggestions. We have carefully rewritten the two methods sections (Sections 3 and 4 in the original manuscript), with the changes highlighted in blue in the updated manuscript. We respond to the individual questions below:
>
> > The probabilistic formulation is missing explanation of the first part of Eq. 2 and 3. The proposed probabilistic formulation seems based on the likelihood depth image model from 3dp3 (Eq. 2) and extended with a weighting factor based on SurfEMB correspondences (Eq. 2).
>
> We have carefully rewritten the explanation for our probabilistic formulation. We have included a new Preliminaries section (Section 3.1 in the updated manuscript), which reviews our noise model on depth information from 3DP3 and our noise model on RGB information from SurfEMB. We have added a detailed paragraph describing our overall mixture formulation (Eq. 3 in the old manuscript, Eq. 1 in the updated manuscript): each observed pixel $(i, j)$ can be explained either by a rendered pixel $(\tilde{i}, \tilde{j})$ which combines noise models on RGB and depth information, or can be explained as a background pixel which is modeled as a uniformation distribution $1/B$ within a bounded volume. In our updated manuscript we also introduce simplified notations (e.g. $P_{background}, P_{depth}, P_{RGB}$) to make the Eq. 1 self-explanatory.
>
> > very difficult to follow the description of the pose hypothesis generation and voting, a visualization of an example could help.
>
> As requested, we have included a new figure (Figure 3 in the updated manuscript) which illustrates both the proposed spherical voting procedure and enumeration-based pose hypotheses generation using a concrete example. We also significantly expanded the description of the pose hypotheses generation and voting process in the updated manuscript, both in the main text (Section 3.3) and in the appendix (Appendix A.3).
>
> > While it seems that the proposed hypothesis generation and spherical voting and scoring improves over SurfEMB initialization it also seems to be improved itself by using additional 2d detections. This raised the question which hypothesis are actually sampled in the optimization and the accuracy of the proposed scoring. A seperate evaluation on the performance of the scoring and success of hypothesis generation for the different methods 1-3 would help to clarify this question.
>
> We distinguish two different setups:
>
> 1. When we do not have access to 2D detections, SurfEMB completely fails, while our proposed hypotheses generation process can robustly aggregate information from the entire image and still produce good results (See 3DNEL MSIGP (No 2D Detection) in Ablations of Table 1).
> 2. Although our proposed hypotheses generation process can robustly aggregate information from the entire image, for many objects the query embeddings are quite noisy. By additionally leveraging 2D detections, we can filter out noise coming from regions outside the object of interest (as determined by the 2D detections) and further improve the quality of the generated pose hypotheses. This allows 3DNEL MSIGP to significantly outperform SurfEMB with exactly the same amount of information.
>
> As a result, the requested evaluation is already included in the paper (the No 2D detection ablation). We believe this confusion is coming from the lack of details on how we use 2D detections in our pipeline. We have rewritten the description on how we use 2D detections in our pipeline (last paragraph on page 6 of the updated manuscript), and also expanded our discussions in the appendix (Appendix A.5) with more details on the use of 2D detections.
>
> > In "Pose hypotheses proposals iterate over the generated pose hypotheses twice and make a proposal centered at each". It is unclear what is meant by this iteration and why it is twice.
>
> Our MCMC finetuning is an iterative process where, at each iteration, we propose to change the pose of a single object using some proposals, and accept the proposal with the MH acceptance probability. Concretely, the pose hypotheses proposals go through the $N$ objects multiple times. For object $i$ of class $t_i$, the pose hypotheses proposals sequentially go through the generated pose hypotheses for object class $t_i$, sample a pose centered at each of the pose hypotheses, and propose to change the pose of object $i$ to the sampled pose. Going through the objects more times would lead to better estimation, at the cost of increased computation. In practice we find going through all objects twice strikes a good balance between speed and performance.
>
> We have rewritten the description on MCMC finetuning (Section 3.3 in the updated manuscript) to make this clearer. In particular, to avoid future confusion, we have changed the main text to say “multiple times”, and describe “twice” in the appendix as a hyper-parameter we choose (Appendix A.4).

---

> ### Author Response · Authors · 2022-11-13
> **Thank you for your constructive feedback (part 2)**
>
> > It also remains unclear how the initial hypothesis are generated. Are they drawn from the uniform distribution?
>
> For positions, we discretize the camera frame space into a voxel grid, and use the centers of the different voxels as our position discretizations.
>
> For rotations, as we describe in appendix A.5, instead of using a uniform distribution, we systematically discretize the rotation space into 6400 rotations. We first pick 200 points on the unit sphere using the Fibonacci sphere (which gives a set of roughly uniformly distributed points on the sphere), and then generate the 6400 orientations by first rotating the axis $(0, 0, 1.0)$ of the object to point to one of the 200 points on the unit sphere, followed by one of $32$ uniformly distributed in-plane rotations around the axis. Compared with random uniform sampling, this more systematic approach can ensure good coverage of the rotation space.
>
> We have updated the enumeration-based pose hypotheses generation section (Section 3.3 in the updated manuscript) with more details and an explicit pointer to the description in appendix A.5.
>
> > How exactly are the 2d detections used then to mask these?
>
> The 2D detections are not directly masking the pose hypotheses. Instead, the 2D detections are used to mask the query embedding images, so that the spherical voting procedure aggregates information only from the regions of the image that are likely relevant to the object (as determined by the 2D detections). Using the 2D detections, we still enumerate and score all the position and rotation discretizations, but discretizations that are far away from the objects are guaranteed to get low scores due to the masking, which allows us to better handle noisy query embedding images.
>
> We have updated our description of how the 2D detection is used (last paragraph of page 6 in the updated manuscript), and also expanded the description of 3DNEL MSIGP in Appendix A.5 with additional details on the use of 2D detection.
>
> > The authors also write "we use the same 2D detection and mask prediction from SurfEMB to obtain query embeddings at the right scale", but it is unclear what is meant by the right scale here.
>
> Here by “scale” we mean the scale we apply to the RGB image to change the size of the RGB image as input to the query embedding model. SurfEMB’s query embedding model was trained on image crops from 2D detections rescaled to $224\times 224$, but can be applied to RGB images of any size. Empirically we observe that due to SurfEMB’s rescaling of the 2D detection crops to $224\times 224$, the query embeddings for some objects are only informative when we properly rescale the input RGB image. To account for this, we estimate the best scale by scoring the top pose hypotheses (using our heuristic scoring as described in Section 3.3) from each 2D detector crop and pick the scale associated with the highest-scoring 2D detector crop.
>
> Note that this setup is using exactly the same information as SurfEMB to ensure a fair comparison. But as we demonstrate in our ablation study (3DNEL MSIGP (No 2D Detection) in Ablations of Table 1), even with just a default upscaling of 1.5x and no 2D detections, we can already achieve highly competitive performance, while SurfEMB completely fails in this case.
>
> To avoid confusion, we have updated the last paragraph on page 6 in the updated manuscript to make it more self-contained, and instead expanded Appendix A.5 to describe the use of this scale estimation in more detail.
>
> > While the generative nature allows for a nice modelling of the probability distribution, unfortunately the modelling and use of the uncertainty is not further investigated.
>
> We thank the reviewers for this valuable comment. As suggested by the reviewer,  we have added a new tracking experiment highlighting the benefit of modeling uncertainty, a key advantage enabled by the probabilistic formulation of 3DNEL. See the general response, the last paragraph in Section 4.3 and Appendix A.7 for a description of the experiment. The experiment clearly demonstrates how proper modeling of uncertainty allows us to handle challenging situations (e.g. tracking objects that get heavily occluded).

---

> ### Author Response · Authors · 2022-11-13
> **Thank you for your constructive feedback (part 3)**
>
> > Unfortunately it would be difficult to reproduce the method from the paper, as some details are unclear e.g. the exact hypothesis generation and sampling as well as hyperparameters e.g. to compare the point and keypoint distances inside algorithm 1.
>
> We have carefully rewritten the hypotheses generation process (Section 3.3 in the updated manuscript), with an explicit pointer to a detailed description in the appendix (Appendix A.3 in the updated manuscript).
>
> All the hyperparameters used in the experiments were already described in the appendix in the original manuscript (Appendix A.5 in the updated manuscript). To avoid confusion, we have added a comment immediately after Eqn. 1 in the updated manuscript, clearly stating that $r$ is a hyperparameter we pick in the experiments, and point to the appendix for more details.
>
> Finally, as mentioned in the paper, upon acceptance we will open source a Python codebase reproducing all the results reported in the paper to ensure reproducibility. We thank you again for the very detailed and constructive comments. Please don’t hesitate to let us know of any additional comments on the paper. We are very happy to address them.

---

### Author Response · Authors · 2022-11-13
**Response to All Reviewers**

We thank all the reviewers for their valuable comments and suggestions. Overall, we appreciate that the reviewers acknowledge our contributions and think that the “direction and ideas of the paper are promising” and “the idea itself is nice and the results show significant improvements” (Reviewer aQXb), our work is “important and interesting to the ICLR community”(Reviewer 4Xvb), and our “performance on YCB-dataset is strong and achieves the state-of-the-art” (Reviewer a254).

### Writing improvements and reproducibility

Reviewers 1 and 2 share some common concerns regarding the clarity of writing on the technical details. To address these concerns, we have carefully rewritten the **entire** methods section and significantly expanded the appendix to improve clarity. In particular, we have

- Added a new figure visualizing the spherical voting and enumeration-based pose hypotheses generation process on a concrete example (Figure 3 in the updated manuscript).
- Carefully rewritten the explanation for our probabilistic formulation (Sections 3.1 and 3.2 in the updated manuscript). We have included a new Preliminaries section, which reviews our noise model on depth information from 3DP3 and our noise model on RGB information from SurfEMB, and completely rewritten the section on evaluating 3DNEL, taking into account reviewer comments and questions.
- Rewritten the descriptions on both the enumeration-based pose hypotheses generation and MCMC finetuning (Section 3.3 in the updated manuscript) to improve clarity, and included additional details in the appendix (Appendix A.3 and A.4 in the updated manuscript).

Please refer to our updated manuscript, with the updates highlighted in blue.

In addition, as already mentioned in the paper, **upon acceptance we will open source a Python codebase reproducing all of the results reported in the paper to ensure reproducibility**.

### Additional experiment highlighting the benefits of modeling uncertainty

We also put together an additional tracking experiment that demonstrates the benefits of **modeling uncertainty**, a key advantage enabled by 3DNEL’s probabilistic formulation.

The experiment concerns object tracking in a synthetic scene containing two objects, with one object being static and the other object moving across the scene. The moving object gets fully occluded by the static object before reappearing.

This is a challenging setup for methods that do not model uncertainty but only rely on a single point estimate, as such methods tend to lose track once the moving object becomes severely occluded. We observe that the current 3DNEL extension to object tracking indeed fails. However, but leveraging MCMC inference to maintain a full posterior distribution on the object poses (represented as multiple particles in a particle filtering setup), we are able to properly model uncertainty and achieve successful tracking: the posterior distribution would spread out across the scene when the moving object becomes highly occluded (since there is a high level of uncertainty on where the object is), which allows the system to continue tracking the object when it reappears.

This experiment is also done using improved infrastructure supporting a large number of 3DNEL evaluations in parallel, and achieves **real-time tracking performance**, although on simpler, synthetic data.

See the added paragraph in Section 4.3 of the updated manuscript and Appendix A.7 for a detailed description. The current version of the experiment only uses the noise model on the depth information to model simple synthetic two-object scenes.. We are additionally putting together a setup using the noise models on both RGB and depth information on more challenging synthetic scenes with YCB objects, and will post the results as soon as we have them ready.

In the individual responses we provide a point-by-point response to the reviewer comments.

---

### Author Response · Authors · 2022-11-18
**Updated tracking experiment with 3DNEL**

We would like to thank all the reviewers again for the time and efforts you put in to provide thoughtful feedback and comments. As promised, we have finished the additional tracking experiment **using the full 3DNEL formulation** to highlight **the benefits of modeling uncertainty**, and we have updated the manuscript accordingly (see Section 4.3 and Appendix A.7 in the updated manuscript).

We have revised the paper according to the your suggestions and replied to all the questions and concerns. Have our responses addressed your comments? We are happy to address any further comments you might have. Thanks a lot!

---

### Decision · Program_Chairs · 2023-01-20

**Decision:**

Reject

**Justification For Why Not Higher Score:**


First, the paper has not put enough effort to compare against those methods, e.g. CozyPose or MegaPose in an apples to apples manner. For example, for CosyPose the code is available and the model could be retrained using only simulated data to have an apples to apples comparison with the proposed method in Table 1. Second, the contribution of the work appears minor, in the sense that the matching function is taken from  previous works. Third, the improved performance of the present method costs 6X the inference time due to gradient-free optimization. In fact, the right comparison plot should also have the inference time on one axis, and methods should be compared both in terms of accuracy but also in terms of inference time efficiency. Forth, given the slow inference of the proposed method, it is unclear why it is disadvantageous to use real world annotations, given that building simulators is also very time consuming, similar  to annotating real world images. The authors are encouraged to use a SOTA previous work, e.g. CosyPose or the most recent MegaPose, and train it in an apples to apples method, as well as show comparisons and contributions in apples-to-apples inference time. Many of previous works would alos benefit from additional inference time by checking more hypotheses, similar to the presented method.   The work is in a great direction, perception as analysis by synthesis, and we encourage the authors to add the suggested comparisons and submit to a future venue.

**Justification For Why Not Lower Score:**

N/A

**Metareview: Summary, Strengths And Weaknesses:**

This paper proposes a method for inference of 3D object pose and locations in RGB-D images and videos. It follows an analysis-by-synthesis paradigm where object pose hypotheses are continuously predicted and evaluated in an MCMC framework. The hypothesis prediction and the evaluation leverages learned 2d-to-3d feature mappings from previous work (SurfEMB) and depth matching from (3DP3). The method is purely trained on synthetic images and evaluated on real data showing some improvements to baselines. During rebuttal time, the authors showed results where the uncertainty provided by their formulation is beneficial for tracking and object 3d pose inference.

There is a long line of work that does 2D to 3D inference or 2.5D to 3D inference similar to this paper:
https://arxiv.org/pdf/2205.15768.pdf and many papers before use what they call a camera multiplex representation that scores multiple camera hypotheses jointly:
 Cosypose  https://github.com/ylabbe/cosypose searches over 3D poses of multiple objects .
 MegaPose: 6D Pose Estimation of Novel Objects via Render & Compare seem to have very similar objectives as the present work,
Most recent work: Learning to Imitate Object Interactions from Internet Videos that again infers 3D object and hand poses via iterative refinement.
First, the paper has not put enough effort to compare against those methods, e.g. CozyPose or MegaPose in an apples to apples manner.  For example, for CosyPose the code is available and the model could be retrained using only simulated data to have an apples to apples comparison with the proposed method in Table 1.
Another important baseline missing whose code is publicly available and shows SOTA performance across multiple benchmarks is: Coupled Iterative Refinement for 6D Multi-Object Pose Estimation.
 Second, the contribution of the work appears minor, in the sense that the matching function is taken from  previous works. Third, the improved performance of the present method costs 6X the inference time due to gradient-free optimization. In fact, the right comparison plot should also have the inference time on one axis, and methods should be compared both in terms of accuracy but also in terms of inference time efficiency. Forth, given the slow inference of the proposed method, it is unclear why it is disadvantageous to use real world annotations, given that building simulators is also very time consuming, similar  to annotating real world images. The authors are encouraged to use a SOTA previous work, e.g. CosyPose or the most recent MegaPose, and train it in an apples to apples method, as well as show comparisons and contributions in apples-to-apples inference time. Many of previous works would alos benefit from additional inference time by checking more hypotheses, similar to the presented method.   The work is in a great direction, perception as analysis by synthesis, and we encourage the authors to add the suggested comparisons and submit to a future venue.

**Summary Of Ac-Reviewer Meeting:**

We concluded that the paper does not place itself well with respect to previous works, in 2D to 3D perception literature.